# A Wholistic View of How Bumetanide Attenuates Autism Spectrum Disorders

**DOI:** 10.3390/cells11152419

**Published:** 2022-08-04

**Authors:** Eric Delpire, Yehezkel Ben-Ari

**Affiliations:** 1Departments of Anesthesiology and Molecular Physiology & Biophysics, Vanderbilt University School of Medicine, Nashville, TN 37232, USA; 2NeuroChlore, Campus Scientifique de Luminy, 163 Route de Luminy, 13273 Marseilles, France

**Keywords:** blood-brain barrier, ASD, neurodevelopmental disorders, parkinson’s disease, alzheimer’s disease, bumetanide, NKCC1, clinical trials, side effects, brain-gut interactions, immune alterations in brain disorders, central and peripheral actions of bumetanide

## Abstract

The specific NKCC1 cotransporter antagonist, bumetanide, attenuates the severity of Autism Spectrum Disorders (ASD), and many neurodevelopmental or neurodegenerative disorders in animal models and clinical trials. However, the pervasive expression of NKCC1 in many cell types throughout the body is thought to challenge the therapeutic efficacy of bumetanide. However, many peripheral functions, including intestinal, metabolic, or vascular, etc., are perturbed in brain disorders contributing to the neurological sequels. Alterations of these functions also increase the incidence of the disorder suggesting complex bidirectional links with the clinical manifestations. We suggest that a more holistic view of ASD and other disorders is warranted to account for the multiple sites impacted by the original intra-uterine insult. From this perspective, large-spectrum active repositioned drugs that act centrally and peripherally might constitute a useful approach to treating these disorders.

## 1. Introduction

Bumetanide (3-(butylamino)-4-phenoxy-5-sulfamoylbenzoic acid) was introduced in clinical medicine in 1972 as a drug presenting higher diuretic potency than furosemide, a compound discovered just eight years earlier [1]. A study published in the *British Medical Journal* reported that in 27 edematous patients, the dosage of bumetanide required only 1/40th of the dosage of furosemide for a similar diuretic effect [2]. The molecular target of bumetanide, furosemide and other related diuretics is NKCC2, a cotransporter of Na^+^, K^+^ and Cl^−^ expressed in the renal thick of ascending loop of Henle [3]. This site of action led to the classification of these drugs as “loop” diuretics, as opposed to thiazide diuretics, which work in more distal segments of the nephron. NKCC2 is a protein/transporter encoded by the solute carrier family 12, member 1 gene, or *SLC12A1*.

Loop diuretics are typically delivered orally or intravenously to treat congestive heart failure and disorders that lead to volume expansion. As with many other drugs, the method of administration greatly affects the time to peak diuretic effect. When taken orally, the diuretic is absorbed by the intestine (Figure 1), thereby delaying the time to peak effect to 60–90 min, compared to the 10–30 min for intravenous administration [4]. Once in the blood, >95% of the diuretic binds to serum albumin, α- and β-globulins, thereby reducing its systemic bioavailability [5,6]. The free diuretic, which is in equilibrium with albumin-bound in the blood, is secreted to the urine by organic transporters where it freely flows to its site of action in the thick ascending limb of Henle. The absence of proteins in the urine guarantees the maximal effect of the inhibitor in the ascending limb. The mechanism of degradation, which is different from drug to drug, also greatly affects the diuretic properties. The majority of furosemide, for instance, is eliminated from the body through urine excretion, whereas the majority of torsemide, another loop diuretic, is eliminated by the liver. It follows that under advanced renal dysfunction, the half-life of furosemide is prolonged, whereas, in conditions of hepatic dysfunction, the half-life of torsemide is doubled. Bumetanide is also partially metabolized by the liver (cleavage of the butyl group) and is excreted in the urine as both drug and metabolite. Its half-life in adults is 1–1.5 h. 

The pharmacokinetic properties of bumetanide and related drugs are critical for proper diuretic effect. They are also important for the minimization of unintended side-effects of the drugs, as at very low concentrations loop diuretics also target NKCC1, a Na-K-2Cl cotransporter widely expressed throughout the body, including the central and peripheral nervous system (Table 1). Here, we discussed the implications of this ubiquitous expression in relation to the therapeutic actions of Bumetanide in Autism Spectrum Disorders (ASD) and other neurodevelopmental disorders. 

## 2. The NKCC1/KCC2 Activity Ratio and GABAergic Inhibition 

NKCC1 and KCC2, best known as chloride importers and extruders, respectively, play a crucial role in maintaining adequate [Cl^−^]_i_ levels. They exert multiple roles such as cell volume regulation, but also define the polarity of the adult major inhibitory transmitter GABA. Indeed, the GABA receptor/channel complex is permeable to anions and in particular chloride—the most abundant anion present. Low [Cl^−^]_i_ levels are indispensable for the generation of the hyperpolarizing/inhibitory actions of GABA, whereas high [Cl^−^]_i_ levels are associated with depolarizing and often excitatory actions of GABA (reviewed in [23]). In physiological conditions, adult neurons of the central nervous system (hereafter referred to as central neurons) maintain low [Cl^−^]_i_ levels due to the high activity of KCC2 respective to the low activity of NKCC1. These low [Cl^−^]_i_ levels are restricted to central neurons, as neurons of the peripheral nervous system (hereafter referred to as peripheral neurons) and other cell types have high [Cl^−^]_i_ levels due to high expression of NKCC1 and the absence of KCC2 [24]. Peripheral sensory neurons, like central neurons, express other isoforms of the K-Cl cotransporter, e.g., KCC3 [25,26], but those are inactive under isosmotic conditions [27]. 

In contrast to their mature counterparts, central immature neurons also have high [Cl^−^]_i_ levels due to strong NKCC1 activity and reduced KCC2 activity. High [Cl^−^]_i_ levels have been observed in all brain structures and animal species investigated extending from worms to primates indicating that it has been preserved throughout evolution [23,28,29]. The resulting depolarizing/excitatory actions of GABA underlie the generation of Giant Depolarizing Potentials (GDPs) a common feature of many immature central networks [30,31]. Excitatory GABA produces a surge of calcium influx through the activation of voltage-gated calcium currents and the removal of the magnesium blockade of NMDA receptors [32]. The calcium surge underlies the trophic role of GABA in brain development modulating cell proliferation, neuronal growth and synapse formation [23,33]. The tightly controlled high [Cl^−^]_i_ levels of immature neurons decrease progressively throughout the birth and postnatal maturation by means of enhancement of KCC2 [34,35] to the detriment of NKCC1 [36]. Collectively, these observations stress the importance of the NKCC1/KCC2 activity ratio in development. 

## 3. The NKCC1/KCC2 Activity Ratio Is Perturbed in Brain Disorders

This GABA polarity switch is altered in a variety of neurodevelopmental and adult degenerative disorders. In a wide range of brain disorders, neurons have high [Cl^−^]_i_ levels and excitatory GABA actions. This includes developmental disorders such as ASD, Rett, Fragile X, some forms of infantile epilepsies, etc.; neurodegenerative disorders such as Parkinson’s, Huntington, chorea; cerebro-vascular infarcts, traumatic injuries, post-traumatic disorder, chronic pain, spinal cord lesions, glioblastoma, severe brain tumors, etc. [37,38,39]. Transcript of KCC2 is reduced in hippocampus of deceased individuals diagnosed with schizophrenia [40]. A recent study also shows that bumetanide is the most effective among 1300 widely used treatments to attenuate APOE genetic signatures of Alzheimer’s disease [41]. The study also shows that bumetanide attenuates physiological and behavioral features of AD in mice. Finally, the use of bumetanide in >65-year-old individuals reduces the incidence of AD by 30 to 70% suggesting that bumetanide might be a promising agent to treat ASD. Collectively these observations suggest that the increased [Cl^−^]_i_ levels are a general feature of pathological conditions and bumetanide remains a good candidate to treat an unprecedented list of disorders. 

Importantly, restoring low [Cl^−^]_i_ levels and therefore inhibitory actions of GABA by either an NKCC1 antagonist or a KCC2 enhancer attenuates the severity of many brain disorders [42,43,44]. This has been extensively illustrated with bumetanide, a specific antagonist of NKCC1, that efficiently restores low [Cl^−^]_i_ levels, GABAergic inhibition and attenuates the severity of disorders (Figure 2). This has been reported in ASD and related neurodevelopmental conditions including Fragile X, Tuberous Sclerosis, Down syndrome, Maternal Immune Activation or Rett syndrome [23,45,46,47,48,49,50]. Bumetanide also attenuated the stress susceptibility produced by maternal-offspring separation [51]. 

Bumetanide also attenuates the severity of ASD in clinical trials extending from pilot trials to double-blind placebo/control trials with an improvement in social interactions and reduced agitation, the children being more “present”. In randomized placebo-controlled trials, a total of 496 children were treated with significant attenuation of ASD [52,53,54]. Visual contact and brain functional imaging have also shown an improvement in face recognition, and activation by emotive figures of brain regions involved in visual interactions [55,56]. Despite these early findings, a final EMA-approved phase 3 trial completed recently (400 children recruited in 45 centers in Europe, the USA, Brazil and Australia) failed to reveal a significant difference between placebo and bumetanide-treated children. However, a recent study shows that bumetanide ameliorates EEG abnormalities present in adolescents with ASD- in parallel with attenuations of the symptoms- and EEG features [57]. In fact, EEG alterations were restored by bumetanide and these effects enable to predict the outcome of bumetanide treatment providing a possible biomarker of ASD (see below). In another trial, cytokine measurement enabled us to identify bumetanide responders in the subpopulation of young children with ASD, relying on blood levels of interleukins [58]. Collectively, these observations suggest that the failure of the trial is due to the need to identify responders. 

Pilot cases have shown an improvement in treating a patient with schizophrenia [59] Fragile X [60] but also Tuberous Sclerosis where bumetanide attenuated ASD symptoms but not seizures [61]. Benzodiazepines exert paradoxical actions in patients with ASD [62], providing indirect evidence that [Cl^−^]_i_ levels might be high in children with ASD. Indeed, in experimental investigations, benzodiazepines increase neuronal activity when the actions of GABA are excitatory [63]. In sum, the convergence of experimental and clinical data reflects the importance of the NKCC1/KCC2 activity and GABA polarity, and particularly bumetanide, as promising treatments of ASD and possibly other brain disorders. They also suggest that the actions of Bumetanide are at least in part mediated by reducing and restoring low [Cl^−^]_i_ levels and GABAergic inhibition. 

At this stage, it might be important to stress the strong convergence of experimental and clinical observations. Thus, in Parkinson’s disease, the inhibitory effects of cortical input to striatal cholinergic neurons—the so-called off response—is abolished in PD and restored by bumetanide. Motor activity is also impaired and restored by bumetanide treatment [64]. Importantly, the *off* response evoked in striatal neurons by cortical stimulation was blocked in dopamine-deprived rodents—because of high (Cl^−^)_i_ and restored by bumetanide [64]. In parallel, bumetanide ameliorates gait in a pilot study of 4 patients [65]. This by no means implies that the actions in rodents and humans are mediated by a similar mechanism, yet this possibility cannot be ignored. However, in spite of the highly reproducible efficacy of bumetanide, the ubiquitous expression and poor penetrability of bumetanide through the blood-brain barrier (BBB) have raised strong concern about its actual site of action. Although the in vivo or in vitro efficacy of bumetanide to restore GABAergic inhibition in pathological neurons has not been challenged, is it possible that the attenuation of ASD and other disorders is entirely due to a peripheral action, the restoration of central GABAergic inhibition being irrelevant to its therapeutic effects? Here, we discuss these issues stressing the continuity of central and peripheral actions and the inadequacy of a complete separation of central and peripheral effects. 

## 4. Brain Accessibility of Bumetanide

One factor limiting brain accessibility that has already been discussed is albumin binding: as most of the compound in plasma is protein-bound, only a fraction of the compound is free and therefore able to diffuse across biological membranes. Intraperitoneal administration of bumetanide in rats results in detectable levels in the brain within minutes, with the diuretic having a half-life slightly longer in the brain compartment than plasma [66]. The levels were 200–300-fold higher in plasma than brain but considering that tissue from one brain hemisphere was homogenized, the concentration in the extracellular fluid might be underestimated. In addition, as the plasma concentration likely included the albumin-bound fraction, the free concentration in CSF versus plasma must be close. Once the bumetanide has crossed the blood-brain barrier, it becomes a substrate for organic ion transporters that ‘actively’ pump it back to the blood [67] in accordance with this poor brain penetrability. Kaila and coworkers have repeatedly challenged the usefulness of bumetanide to treat brain disorders [68,69]. However, in a recent study, this group showed that parenteral bumetanide decreases brain inflammation, whereas direct central administration aggravated it, indicating that enough bumetanide reached the brain—notably glia—to exert its effects [19].

The BBB like many other biological barriers is not an absolute static parameter but is dynamic and affected during disease states. It is principally formed by junctions between endothelial cells, structures that restrict the diffusion of solutes between blood and cerebrospinal or interstitial fluids. The barrier forms during embryonic development. Although it is generally believed that the barrier is leakier in newborns, new evidence shows that many characteristics of the adult BBB exist early in development [70]. Tight junctions for instance are well-formed in the early stages of vascularization of the brain [71]. This new recognition does not preclude fetal/newborns-specific characteristics (e.g., strap junctions between neuro-ependymal cells) or adult-specific characteristics (e.g., free diffusion between ependymal cells) of the BBB barrier. A significant difference between youth and adults might be astrocytes, which are integral to adult BBB function, and which differentiate and encircle capillaries in the first 3 weeks of postnatal life in rodents [72]. Thus, significant functional differences might exist between the brain during its latest phases of development and the adult brain. In addition, little data is available in the literature regarding the BBB permeability in humans and in the pediatric population in particular. In the adult, the BBB protects well the nervous system from unwarranted toxins that circulate in the blood. However, a reduction in BBB effectiveness has been reported in many central and peripheral disorders. These include inflammatory states such as diabetes [73,74], rheumatoid arthritis [75,76], cancer [77], etc. Interestingly, some of these detrimental effects of inflammation on BBB can also be improved or reversed by exercise. This was shown both in rats and men [78,79]. Importantly, the blood-brain barrier seems to also break down in Alzheimer’s disease and other neurological conditions including chronic traumatic encephalopathy, amyotrophic lateral sclerosis, Huntington’s disease, multiple sclerosis, or IHV-associated dementia [80,81]. Autistic patients have an elevated immune response with the release of cytokines such as TNFα, IL6 and many others [82,83], this leading to the disruption of the blood-brain barrier [84]. Immaturity of the BBB has also been suggested as a possible factor in ASD [85]. Post-mortem studies suggest that the BBB and the intestinal epithelial barrier are altered in patients with ASD and related disorders [86]. Thus, it is not unreasonable to consider the possibility that bumetanide might better penetrate the brain under these pathological conditions. 

Again, it bears stressing that the majority of bumetanide concentration values reported in the literature are from healthy rodent brains and it remains unclear whether these values are valid for children, especially those with ASD. Therefore, if bumetanide is shown to exert positive therapeutic actions, in spite of this limitation, it deserves to be tested in clinical trials to treat disorders with at present unmet therapies. Rejecting that on the basis of rodent measures of BBB is ethically debatable. The ubiquitous presence of NKCC1 raises the question of the roles of peripheral actions of bumetanide: is this presence deleterious thereby challenging the “primum non nocere” doctrine that is required by clinical authorities; or are they part of effects observed contributing somehow to the clinical amelioration produced by bumetanide? The first possibility is unlikely in view of the medically approved long-lasting use of bumetanide. Let us examine the side effects of bumetanide in relation to other widely used treatments. 

## 5. Peripheral Actions 

### 5.1. Detrimental Effects on Inner Ear

Side effects of bumetanide have been extensively studied over the 6 decades of their wide use to treat volume expansion states such as edemas and hypertension. The renal side effects of bumetanide are restricted to diuresis, dehydration and possible blood hypokalemia, which is readily controlled by supplementing the patient diet with food rich in potassium. Ototoxicity has raised some concerns, after being observed in newborns suffering from severe encephalopathy, e.g., in the often quoted trial of Pressler and colleagues [87] which treated 2-day-old babies affected by encephalopathy with bumetanide (4 i.v. injections), antiepileptics (phenobarbital, midazolam, phenytoin and lidocaine) and antibiotics (tobramycin or gentamycin) which are known to have ototoxic effects [88]. Ototoxicity has never been observed in hundreds of children treated orally with bumetanide, even after several years. Ototoxicity has been occasionally reported in adolescents or adults however only after intravenous administration of high dosages, but never to the best of our knowledge, after oral administration. This includes the doses used in most pediatric clinical trials on ASD (0.5 to 2 mg twice daily). In adults, diuretic ototoxicity is principally observed in patients with diminished renal function, that is patients who have difficulties eliminating the diuretic [89,90]. Otherwise, bumetanide is safe and even long-term treatment with the diuretic elicits only minimal side effects [91]. Why are loop diuretics affecting the inner ear? The answer resides in the expression of NKCC1 in the stria vascularis, a multi-layer epithelium involved in the production of the K^+^-rich endolymph and in the formation of the cochlear potential [92]. In agreement with the role of the cotransporter in the inner ear, the complete absence of NKCC1 function in both mice and humans leads to a deficit in endolymph production and profound deafness [7,8,93,94]. Interestingly, single allele mutations in NKCC1 have also been associated with hearing loss [95,96], while a reduction in cotransporter activity is involved in age-related deafness [97]. Out of all loop diuretics, bumetanide is in fact the safest, as it is 6–9 times less effective in producing inner ear toxicity than ethacrynic acid and furosemide at equivalent diuretic dosage. This observation was made in cats [98], dogs [99] and humans [100]. Thus, low doses of bumetanide are perfectly safe for inner ear function, as now established by a very large number of adult and pediatric patients that have taken bumetanide.

Children and adolescents with ASD have important auditive excessive sensitivity and hyperacusis [101]. This is associated with abnormalities of the brain stem in children with ASD [102,103]. Interestingly, the developing inner cochlea follows the GABA excitatory/inhibitory developmental shift with a reduction of the NKCC1 chloride importer and depolarizing actions of GABA that activate voltage-gated calcium channels [104] and GABA exerts depolarizing actions early on as in other brain structures [105]. These alterations may be part of a generalized process that underlies the heterogeneity of the syndrome [106] with a convergence of central and peripheral effects that involves NKCC1.

### 5.2. CSF Production

Experiments performed some 50 years ago sought to determine if furosemide, a potent diuretic that abolishes the renal medullary sodium gradient, affected sodium uptake in choroid plexus, the epithelium in the brain that produces 55–75% of the cerebrospinal fluid [107,108]. They noted that in rabbits, furosemide at 10 mg/kg, i.v., produced little if any effect on CSF flow, while 50 mg/kg, i.v., reduced CSF flow by 45% [109]. Similarly, Domer reported that 20 mg/kg furosemide, administered i.v. in cats, also did not produce a significant alteration in the rate of fluid formation [110]. To circumvent the barrier, Buhrley and Reed then administered the drug intraventricularly and observed a reduction in CSF secretion [111]. In their paper, they explained the lack of i.v. effect as follows: “*It is possible that the reported lack of effect could be attributed to an insufficient concentration of the agent at the site of action. This, in turn, would be a function of the dose and of the route of administration of the drug. It is possible that most of the drug may have been excluded from the site of action after intravenous administration*”. What they had done in their two papers is to provide the first evidence for the apical localization of the Na-K-2Cl cotransporter in choroid plexus epithelial cells, which was later ignored as most models placed the cotransporter on the basolateral membrane, mostly for convenience. Apical localization was re-introduced in 1994 by Richard Keep [112] and confirmed with antibodies developed against NKCC1 [36,113]. The fact that NKCC1 is involved in CSF production is clearly established but the mechanism is still controversial [114,115]. Similarly, loop diuretics reduce intracranial pressure by inhibiting CSF production in cats [116], dogs and baboons [117]. preterm and term rabbits [118,119]; and reduce intraventricular pressure in patients undergoing neurological surgery [120,121]. These effects might be beneficial from a therapeutic perspective. 

### 5.3. Vasculature

Effects of diuretics on the vasculature are relatively difficult to ascertain due to the volume contraction induced by the diuresis. Several studies have however shown that NKCC1 activity influences vascular smooth muscle tone [122]. Vasoconstrictive agents, such as angiotensin II, phenylephrine and endothelin increase NKCC1 activity, whereas agents causing vasodilation, e.g., nitric oxide and nitroprusside, inhibits the transporter [122]. In agreement with a role of NKCC1 in vasoconstriction, the NKCC1 knockout mouse is hypotensive [8,123]. Another interesting observation in this context arose from experiments performed in wild-type mice showing that intravenous infusion of bumetanide produced an immediate drop in blood pressure, not prevented when the renal arteries were clamped, which clearly indicated that the bumetanide-induced drop in BP was not kidney mediated [17]. Thus, NKCC1 inhibition leads to vasodilation, which if that involves cerebral blood vessels will lead to increased cerebral blood flow.

Bumetanide-like furosemide exerts a wide range of effects on both peripheral and central sites including the microvascular central system [124]. Bumetanide exerts via this effect a protective action on ischemic brain damage [125]. Therefore, the reported protective actions of NKCC1 antagonists might well be due to a convergence of central and peripheral effects.

### 5.4. Immune System

Cells from the immune system are activated by a variety of conditions which include not only acute response to pathogen invasion but also many chronic disease conditions such as diabetes, cancer, autoimmune diseases and neurological disorders, etc. The function of immune cells is complex and modulated by many factors that activate, amplify, repress, or terminate the immune response. Ion channels and transporters are included in these processes. NKCC1, again likely through the accumulation of intracellular Cl^−^, affects the activity of immune cells. Experiments performed in the mid-1990s showed that activation of respiratory burst, production of reactive oxygen species, adhesion and spreading of human neutrophils were triggered by Cl^−^ efflux [126]. However, this process was not affected by pretreating with furosemide, an inhibitor of NKCC1. In contrast, bumetanide attenuated the activation and production of inflammatory cytokines of macrophages stimulated by lipopolysaccharides (LPS) [20]. The same study showed that mice treated with an intratracheal application of bumetanide showed greater resistance to LPS-induced tissue inflammation and acute lung injury. The process of macrophage efferocytosis, which is the engulfment of dead cells or bacteria, is also largely modulated by intracellular Cl^−^ as it is accelerated by deletion of the Cl^−^ importer NKCC1, while reduced by deletion of the Cl^−^ exporter, KCC1 [21]. In the brain, inflammation stimulates the secretion of cerebrospinal fluid by the choroid plexus through Toll-like receptor 4. NFKB, SPAK kinase and NKCC1 activity [127]. This process is eliminated with the intraventricular application of bumetanide. In the same vein, systemic bumetanide treatment prevented the release of proinflammatory cytokines by microglia in a rat model of postsurgical brain injury [128]. Epigenetic analyses of individuals with ASD or attention-deficit/hyperactivity disorder and Tourette syndrome demonstrate convergent dysregulated immune pathways [50]. Collectively, the attenuation of inflammation and activation of the immune system are compatible with a positive effect of bumetanide on pathogenic events involved in brain disorders. 

Bacterial or viral infections during pregnancy like the familial history of autoimmune disease increase the incidence of brain disorders notably ASD [129,130]. In experimental conditions, Maternal immune activation (MIA) is linked to ASD and schizophrenia [131,132,133]. MIA yields offspring displaying the main core symptoms of ASD [134] and delays the GABA excitatory to inhibitory developmental shift [135]. This alteration of the GABA shift has been observed in a wide range of disorders including Fragile X, Down syndrome, ASD, Rett syndrome and neurodegenerative disorders [37,39]. Bumetanide restores GABA polarity and attenuates the behavioral signs of ASD (ibid). 

### 5.5. Enteric System

The Microbiota/Gut/Brain axis has been extensively investigated [136]. Gut microbiota plays a key role in the life and health of the host by protecting against pathogens, and beyond the gastrointestinal tract plays a major role in the bidirectional communication between the environment and the brain. In addition to parallel developmental processes between the gut and neurons, disruptions in gut microbiota-host interactions profoundly impact brain-gut signaling and increase the risks of developmental disorders. Gut-brain interactions take place via the immune system, alterations of microbiota or neural pathways via the vagus nerve that is directly impacted by microbiota [134]. Like central immature neurons, GABA exerts depolarizing actions on the enteric system notably the colon [137]. The secretion of the incretin-glucagon-like peptide triggered by food intake is also sensitive to GABA (and Glycine) via bumetanide-sensitive NKCC1 inhibition [138]. Several neurobehavioral, neurodegenerative, mental and metabolic disorders, including Parkinson’s disease, Autism Spectrum Disorder, schizophrenia, Alzheimer’s disease, depression and obesity, have been linked to the gut microbiota [136,139,140]. Collectively these studies illustrate the importance of the bidirectional dual gut/brain axis and its links with brain disorders and with chloride co-transporters. NKCC1 antagonists can attenuate insults in both systems. 

## 6. Conclusions—Brain and Peripheral Organs: A Continuum

In summary, the peripheral actions of bumetanide do not handicap its clinical use and might in fact take part in the attenuation produced by NKCC1 antagonists. The separation of central and peripheral actions is artificial when referring to complex biological functions and a wide range of peripheral and central disorders. ASDs are not restricted to central sequels. Thus, there are numerous interactions between gut microbiota, inflammatory responses and the immune system that impacts behavior and is altered in ASD [131,141]. Gut microbiota is altered in children with ASD [142]. A dysbiosis of the microbiota may influence the development and severity of ASD [132]. Dysregulation of innate immune response and neuronal activity-dependent genes is also observed in ASD [143]. An increased abundance and diversity of *Clostridia* species and a general increase in non-spore-forming anaerobes and microaerophilic bacteria have been reported in comparison to neurotypical controls [144]. A gut-brain neural circuit links intestine epithelial cells with central neurons through a single synapse. Gut epithelial cell synapse on the vagus nerve, transporting information centrally within milliseconds [145]. Autoantibodies against brain epitopes in mothers of children with ASD and children with ASD indicate an aberrant immune response and a disruption of the Blood-Brain Barrier [84]. Epidemiological studies show a strong correlation between maternal or infantile atopic diseases such as food intolerance or allergies, asthma and eczema with risk of ASD [84]. These are mediated by mast cells located in the brain including the thalamus and hypothalamus that increase BBB permeability and regulate emotions (ibid). Transcriptomes of human cortical tissue samples of patients with ASD have identified gene clusters associated with increased microglia activation [143]. Actions on central neurons not protected by the BBB also might contribute to the effects of bumetanide. Thus, GABA and the GABA agonist muscimol stimulate the release of adrenocorticotropic hormone (ACTH) or the growth hormone (GH) in pituitary cells. Consistent with a depolarizing action of GABA, RT-PCR analysis from cultured anterior pituitary cells (obtained from adult female rats), unveiled high levels of NKCC1 but not KCC2 mRNA [146]. Bumetanide may act on these cells by reducing serum levels of ACTH, GH and cortisol which in autistic subjects were found to be higher as compared to controls [147]. These and other observations on the interactions between peripheral and central alterations in ASD and other brain disorders reflect the inadequacy of separating completely peripheral and central actions of treatments. The peripheral and central actions of NKCC1 antagonists might on the contrary converge to reinforce their therapeutic effects. Clearly, the common neuronal response to such a variety of insults constitutes an alternative to the primarily genetic-based strategy targeting the initial insult. A revision of our efforts to develop novel treatments is clearly warranted.

## Figures and Tables

**Figure 1 cells-11-02419-f001:**
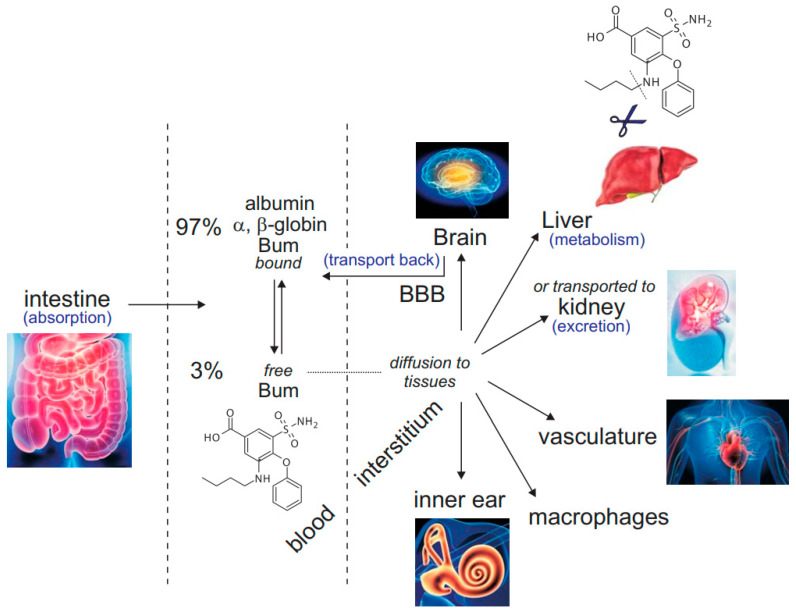
Bumetanide after oral administration. Once absorbed by the intestine, the inhibitor exists in equilibrium as free (3%) versus protein-bound bumetanide (97%). From the blood compartment, bumetanide is transported to the kidney nephrons by specialized transporters where it acts in the pro-urine on the thick ascending limb of Henle to inhibit NKCC2-mediated Na^+^ reabsorption. It is then excreted into the bladder. Bumetanide also diffuses to other tissues where it can act as an NKCC1 inhibitor, provided that its concentration in the periphery reaches values high enough to affect the cotransporter. This includes vasculature, immune cells, inner ear, etc. In the liver, bumetanide is cleaved and metabolized. Because the molecule is rather hydrophobic, it likely crosses the blood-brain barrier (BBB) but is likely transported back to the blood compartment.

**Figure 2 cells-11-02419-f002:**
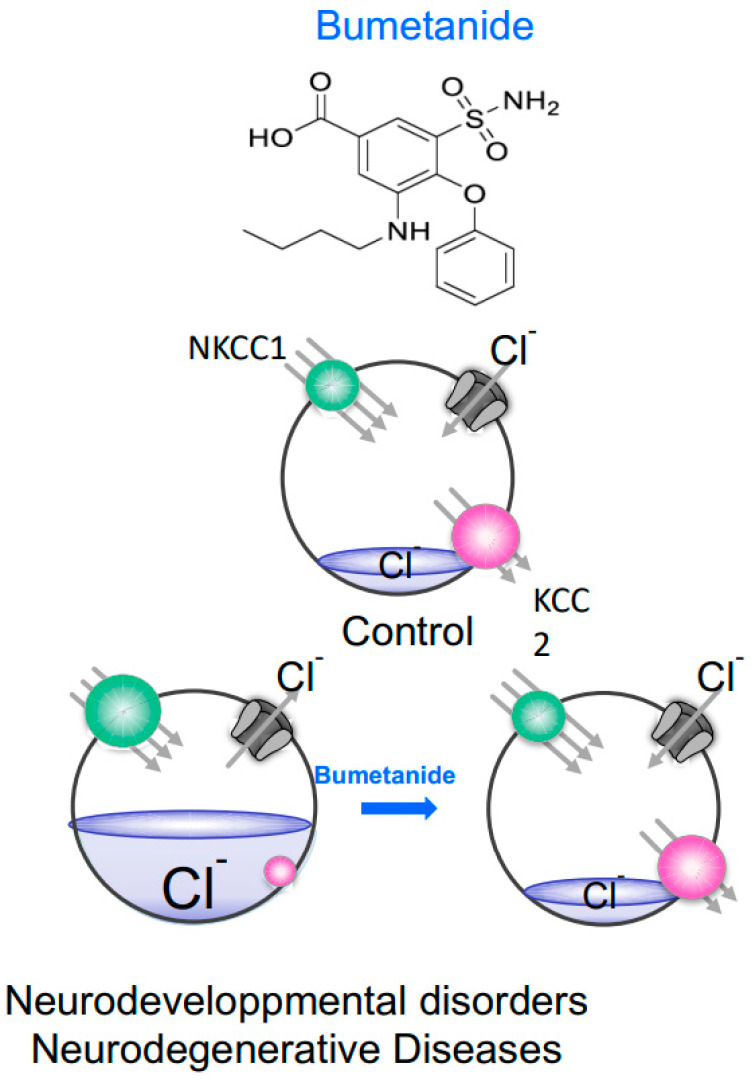
Neurons under control conditions, neurodevelopmental disorders and/or neurodegenerative diseases. In control mature neurons, the activity of KCC2 (red) dominates over NKCC1 (green) and intracellular Cl^−^ is actively kept low. When the GABA receptor (grey) is activated, Cl^−^ enters the cell, which hyperpolarizes the membrane. In neurodevelopmental disorder or neurodegenerative disease, NKCC1 function is stimulated whereas KCC2 function is diminished and as a result, intracellular Cl^−^ increases returning the neuron to an immature state with depolarizing GABA responses. In these cases, bumetanide by inhibiting NKCC1 facilitates the return to low Cl^−^ concentrations.

**Table 1 cells-11-02419-t001:** Expression and Function of NKCC1 in the periphery.

Tissue	Cell Type and Polarity	Function	References
Inner ear	stria vascularis	hearing, balance	[7,8]
Lung	alveolar basolateral	fluid secretion hydration	[9]
Stomach	epithelium basolateral	fluid secretion	[10]
Intestine	epithelium basolateral	Cl^−^ and fluid secretion	[11]
Kidney	Glomerulus, A-type intercalated, IMCD	acid and fluid secretion	[12,13]
Salivary gland	epithelium basolateral	saliva secretion	[14,15]
Sweat/lacrimal gland	epithelium basolateral	sweat and tear secretion	
Skeletal muscle	myocyte	facilitates depolarization, Ca^2+^ entry and contraction	[16]
Vasculature	Smooth mucle cell	facilitates depolarization, Ca^2+^ entry and contraction	[17,18]
Immune cells	Macrophages, Microglia	activation	[19,20,21]
Testis	spermatogonia	sperm production	[22]

## Data Availability

The study did not report any data else than published already.

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
