# Peer review of "A Wholistic View of How Bumetanide Attenuates Autism Spectrum Disorders"

_cells, 2022, doi:10.3390/cells11152419_

Round 1
Reviewer 1 Report
This is a timely and comprehensive review article on the potential pharmacological treatment of ASD. Both authors are experts in the field of chloride transporters and their functional role in developmental neurobiology. Of central interest here is NKCC1 and bumetanide. Although a recent phase 3 trial failed in demonstrating that bumetanide has a significant effect in treating ASD, a number of key issues are still unsolved (e.g. BBB penetration of bumetanide under pathological conditions). This article summarizes these open questions and provides an overview on future perspectives of novel strategies to treat ASD.
I have only very few comments.
1) The authors may want to consider to include the following references.
Hu D, Yu ZL, Zhang Y, Han Y, Zhang W, et al. 2017. Bumetanide treatment during early development rescues maternal separation-induced susceptibility to stress. Sci. Rep 7: 11878
Zimmerman AW, Connors SL. 2014. Could Autism Be Treated Prenatally? Science 343: 620-21
2) I found a few typos:
Page 6: Write "… levels and GABAergic inhibition."
Page 9: Write "… in mid 1990s showed …"
Page 9: Write "… that penetrate through the BBB or not …"
Reviewer 2 Report
The manuscript „A wholistic view to treat Autism Spectrum Disorders” presents an interesting point of view on potential ASD treatment. Bumetanide is tested as potential medicine in ASD for several years and there is no unequivocal conclusion concerning its beneficial action.
In the presented manuscript authors concentrated on the effects of bumetanide and tried to show that its effects in ASD are more complex than only a direct effect on the brain. However, in my opinion, the manuscript is more like a paper describing peripheral actions of this drug with only the symbolic attempt at linking them with ASD. For example, the connections between the enteric nervous system and central nervous system and the action of bumetanide might be interesting. Moreover, following the thesis that the control of chloride ions in the brain is one of the effects of improving ASD conditions, why the authors did not cite research where other compounds (more BBB penetrable) acting on NKCC1 were used.
Anyway, the manuscript does not present what it promises in the title.
Minor comments:
1. The authors describe mechanisms of degradation of furosemide and torsemide; how is it with bumetanide?
2. The phrase “central neurons” is rather unfortunate. It is better to use the common “central nervous system” or simple “neurons”.
3. The authors declare to describe data concerning ASD and neurodevelopmental disorders but much information concern more neurodegenerative than neurodevelopmental disorders.
4. Page 6 – what paradoxical actions of benzodiazepines do the authors mean? “
5. Page 6 – “… reducing restoring low [Cl-]I & GABAergic inhibition.” – is it what the authors wanted to write? Please use “and”.
6. Multiple type and grammar mistakes.
Round 2
Reviewer 2 Report
„A wholistic view to treat Autism Spectrum Disorders” is a well-written and interesting manuscript. However, it does not present what it promises in the title. Looking at the title the reader expects information concerning the spectrum of therapies used in ASD, both non-pharmacological and pharmacological. Meanwhile, the information provided in the manuscript relates mainly to the central and peripheral actions of bumetanide in ASD and other neurodevelopmental (and also neurodegenerative) diseases. This is in fact a holistic view of bumetanide. Therefore I suggest changing the title to one that tells that this is a wholistic view of how bumetanide works in ASD therapy.
I hope that the authors will agree with me. I do not see any other obstacles in publishing the manuscript if the authors agree to such a small change.
Author Response
I agree with the referee
title
a wholistic view of how bumetanide attenuates Autism Spectrum Disorders